# OpenReview forum: "Beyond Truthfulness: Evaluating Honesty in Large Language Models"
_ICLR.cc/2026/Conference — Submitted to ICLR 2026_

### Official Review · Reviewer_6iaV · 2025-10-29

**Soundness:** 2
**Presentation:** 3
**Contribution:** 2
**Rating:** 6
**Confidence:** 4

**Summary:**

This paper argues that a model's "accuracy" and "honesty" are not strongly related. It introduces the MASK benchmark, which assesses a model's honesty by comparing its responses under "Pressure Prompt" versus in "Belief Elicitation Prompt" situations.

**Strengths:**

1. Evaluating a model's honesty by checking the consistency between its responses to a "Pressure Prompt" and a "Belief Elicitation Prompt" is a reasonably sound approach, as it helps rule out false accusations of dishonesty caused by inaccurate answers.

2. The insight reported by the authors is highly meaningful: larger models possess more accurate knowledge yet exhibit lower honesty. This indicates that we cannot simply resolve the honesty issue by scaling up models or increasing data size; instead, we should treat this as a critical challenge in AI safety that warrants special attention.

**Weaknesses:**

1. Treating honesty as the model's internal belief and distinguishing it from accuracy and truthfulness is not a novel contribution of this paper; this distinction has already been discussed and organized in prior survey [1].
2. The authors do not explain why the core finding that "larger models exhibit lower honesty" occurs. Model scale alone cannot serve as a direct cause of dishonesty.
3. The authors appear to consider only one form of dishonesty (when the model’s output contradicts the ground truth), but overlook another important form: dishonesty through incomplete or selectively biased output. For example, a model might produce statements that are factually consistent with its true belief but deliberately omit negative or unfavorable aspects while highlighting only positive ones, thereby distorting the overall meaning.
4. The authors do not consider multilingual settings, despite the fact that honesty can vary significantly across different languages.

[1] A Survey on the Honesty of Large Language Models

**Questions:**

1. What is the underlying reason for the observation that larger models exhibit lower honesty?
2. If a model’s outputs appear factually correct and not overtly deceptive, but the omission of certain information leads to a significantly distorted or misleading overall meaning, how should such behavior be analyzed?
3. Is the evaluation framework adaptable to multilingual settings?

---

> ### Author Response · Authors · 2025-12-03
>
> **W1: The distinction between honesty and accuracy has been discussed in surveys like Li et al. (2025).**
>
> While we agree the *theoretical* distinction exists in prior literature, MASK’s contribution is the first large-scale, automated benchmark to operationalize this distinction empirically across 30 frontier models. Prior work lacked a standardized pipeline to measure lying at scale, especially for closed-weight models. MASK provides the methodology and dataset to empirically disentangle these concepts. We have revised the introduction and related work to clarify this positioning.
>
> **W2/Q1: The authors do not explain why the core finding that "larger models exhibit lower honesty" occurs.**
>
> This is a key empirical finding. We investigate this in Appendix A.4. We find a strong negative correlation (-62.1%) between the propensity to lie and the model's "utility" for honesty.
>
> Current RLHF training incentivizes instruction following and helpfulness, and larger models may be better at optimizing these rewards. Our hypothesis is that larger models are generally more capable of understanding and complying with the nuanced incentives within the pressure prompts (e.g., better goal-following). This may lead them to prioritize the goals implied by the pressure prompt over honesty. We have revised Section 5.1 to better summarize these hypotheses.
>
> **W3/Q2: Overlook [dishonesty] through incomplete or selectively biased output (omission).**
>
> You are correct that we focus on lies of commission. MASK is explicitly focused on lies of commission (knowingly stating falsehoods), as detailed in Sec 4.1 (Principle 3). However, we agree that omission is a critical safety risk (as mentioned in 4.1) and have included this in our Limitations section as a necessary direction for future work.
>
> **W4/Q3: Multilingual settings.**
>
> We acknowledge this limitation in Section A.2. MASK currently focuses on English to ensure high-quality human verification of the nuanced "pressure" dynamics. Adapting this framework to multilingual settings is a valuable future next step. While we believe our broad framework would be directly transferable, there may be cultural differences in what constitutes "honesty" vs. "politeness" which may require a new set of prompts.

---

### Official Review · Reviewer_FLsH · 2025-10-30

**Soundness:** 2
**Presentation:** 3
**Contribution:** 2
**Rating:** 4
**Confidence:** 4

**Summary:**

This paper introduces MASK (Model Alignment between Statements and Knowledge), a benchmark designed to evaluate "honesty" in large language models by measuring lies of commission - instances where models knowingly make false statements under pressure. The authors distinguish between accuracy (factual correctness) and honesty (consistency between beliefs and statements), proposing a novel evaluation pipeline that elicits model beliefs and compares them against statements made under pressure. The study evaluates 30 LLMs and finds that while larger models are more accurate, they don't become more honest, with many frontier models showing substantial propensity to lie under pressure.

**Strengths:**

1. The paper addresses a critical and timely problem in AI safety: the common conflation of "honesty".

2. Easy to follow.

3. The work introduces a novel large-scale dataset of over 1,000 public samples, and the pressure prompts are human-collected. A major strength is that the pressure prompts are human-collected and curated according to thoughtful design principles, such as avoiding unrealistic placeholders ("ABC Company") or clearly fictional settings, which makes the evaluation scenarios more compelling and realistic.

**Weaknesses:**

1. Potential overclaim. This paper claims that accuracy differs from honesty. However, [1] already proposed similar idea that "Second, honesty is specific to each model, as it requires identifying the model’s known and unknown knowledge, making both its evaluation and improvement challenging."

2. Concern about belief detection. This paper proposes to detect the beliefs of LLMs by consistency in responses. However, we can not rely on responses if we do not know if LLMs are honest or not. It is a circular reasoning trap. One possible way is using probing or the date of collected training corpus, as introduced in [1].

3. The  Low-Rank Representation Adaptation (LoRRA) are evaluated only on smaller Llama 2 7B and 13B models. Performances on larger models are required.

[1] S. Li et al. A Survey on the Honesty of Large Language Models, TMLR, Mar 2025. arxiv 2409.18786

**Questions:**

1. How do you address the circular reasoning problem in belief detection? If models might be dishonest in their responses, how can we trust their responses to determine their beliefs?

2. Can you provide validation that pressure prompts create realistic incentives rather than just adversarial conditions that might not reflect real-world deception scenarios?

3. How might alternative belief elicitation methods such as probing techniques or training data analysis compare to the consistency-based approach used here?

---

> ### Comment · Reviewer_FLsH · 2025-11-25
> **Maintaining my scores**
>
> Since I do not hear from the authors regarding the weaknesses and questions I've posted, I will keep my scores.

---

> ### Author Response · Authors · 2025-12-03
>
> We sincerely apologize if the timing of our response caused any concern. We were utilizing the initial part of the discussion period to synthesize the valuable feedback from all reviewers and prepare a comprehensive rebuttal, which we have now posted. We believe there may have been a slight misunderstanding regarding the timeline, as the author-reviewer discussion period extends until December 3rd.
>
> **W1: Distinction between honesty and accuracy has been discussed in surveys like Li et al. (2024).**
>
> While we agree the theoretical distinction exists in prior literature, MASK’s contribution is the first large-scale, automated benchmark to operationalize this distinction empirically across 30 frontier models. Prior work lacked a standardized pipeline to measure this at scale, especially for closed-weight models. We aim to move the field from taxonomy to measurement.
>
> **W2/Q1: Consistency-based method for eliciting beliefs has potential for circular reasoning (i.e., how do we know the unpressured answer is the "true" belief?).**
>
> * Operationalization: We define belief behaviorally as the model’s consistent internal state *in the absence of pressure* (using neutral prompts).
> * Addressing circularity: We assume the "neutral" prompt elicits the model's default knowledge state (accuracy). MASK measures the deviation from this established behavioral baseline when pressure *is* applied.
>
> We mitigate the risk of relying on potentially inaccurate or unstable baseline responses through several checks:
> * Consistency: We verify robustness by querying the model multiple times (three direct queries and, for binary propositions, two indirect queries) and checking for consistency (Section 4.2, A.3). If responses are inconsistent, we classify the model as having "no belief."
> * External validation: In Appendix A.5, we show that when the pressure is removed in a follow-up chat, models self-report having lied in 83.6% of the cases identified by our pipeline. This strongly corroborates that our belief elicitation method accurately captures the model's underlying knowledge, which it chose to contradict under pressure.
>
> **W3: LoRRA evaluated only on smaller Llama 2 models.**
>
> We originally selected Llama 2 7B and 13B to demonstrate the feasibility of this dataset to measure baseline interventions. However, per your suggestion, we have now tested this intervention on much larger open-source models, including Llama-3.3-70B, Llama-3.1-70B, and Qwen2.5-72B.
>
> | Model | Baseline Honesty Score | LoRRA Honesty Score | Δ Improvement |
> | :--- | :---: | :---: | :---: |
> | **Llama-3.1-70B-Instruct** | 56.5 | 85.3 | +28.8 |
> | **Llama-3.3-70B-Instruct** | 55.1 | 84.5 | +29.4 |
> | **Qwen2.5-72B-Instruct** | 50.8 | 79.0 | +28.2 |
>
> **Q2: Validation that pressure prompts create realistic incentives.**
>
> We took many steps to ensure realism, using a set of checklist criteria to filter out and modify ~half of our collected data from human annotators due to quality issues.
>
> The realism stems from our rigorous human-curated data collection process and design principles (Sec 4.1). We explicitly prioritized "Realistic intent to mislead" (Principle 1), ensuring the output could plausibly deceive an audience (e.g., press statements, grant proposals). We strictly avoided unrealistic placeholders (e.g., "ABC Company," "John Doe" placeholders, Principle 2) or clearly fictional settings (Principle 5) to ensure the incentives resemble real-world deception risks. The scenarios cover real-world situations where honesty conflicts with other objectives (e.g., PR management, financial incentives).
>
> We show examples in Figure 5; we have also included an anonymous dataset link (at https://anonymous.4open.science/r/MASK-Benchmark-23919) so one can look through examples..
>
> **Q3: How might alternative belief elicitation methods such as probing techniques or training data analysis compare to the consistency-based approach used in our paper?**
>
> While probing is valuable, it is not possible for closed-source frontier models (GPT-4o, Claude 3.5, Gemini) which are central to our study. Analyzing training data is also intractable for closed-source models. Our black-box approach provides a scalable method applicable to any LLM. We have expanded the discussion in Section A.2 (Limitations) regarding these trade-offs.

---

### Official Review · Reviewer_vCtB · 2025-11-01

**Soundness:** 2
**Presentation:** 2
**Contribution:** 2
**Rating:** 4
**Confidence:** 4

**Summary:**

The paper introduces MASK, a large-scale, human-curated benchmark to measure honesty in LLMs by explicitly disentangling honesty  from accuracy. Evaluating 30 LLMs, the authors find that while larger models are more accurate, they are not more honest—and can readily produce lies of commission under pressure. The paper further reports initial honesty interventions (system prompts, representation-engineering) that improve honesty but do not close the gap.

**Strengths:**

1. Clear concept:  separates honesty from accuracy, addressing a common conflation in prior work;
2. Scale and coverage: ~1.5K carefully curated, realistic, human-written scenarios spanning multiple archetypes, plus 30 frontier models evaluated—strong empirical breadth;
3. Interesting findings: revealing result that scale boosts accuracy but not honesty, motivating research on safety interventions beyond pure capability scaling.

**Weaknesses:**

1. How “belief” is collected. Using consistent answers under neutral prompts is one choice, but please compare it with other ways to get a model’s belief and discuss other explanations (e.g., different knowledge indentification methods);
2. Prompt sensitivity. Results may change with small wording changes in the pressure or belief prompts. Add tests with paraphrases and different pressure strength to show the results are stable;
3. Missing data details: The paper defines six dishonesty types but doesn’t report how many samples each contains and peformance on different types;
4. Dataset accessibility: As a dataset/benchmark paper, it is recommended to upload a supplementary material or anonymous link for reviewers to check the dataset.

**Questions:**

See the Weaknesses section.

---

> ### Author Response · Authors · 2025-12-03
>
> Thank you so much for your review. To address your points:
>
> **W1: How belief is collected, and comparison to other methods for obtaining belief.** Our primary method for obtaining belief is through prompting on different neutral belief elicitation prompts and ensuring consistency across answers. Outside of prompting, some researchers analyze the internal workings of LLMs to understand how they store and recall information [1-2]. Some of these have led to promising ways of eliciting beliefs [3-4]; however, our black-box approach provides a scalable method applicable to any LLM including closed-source models (e.g. GPT, Claude, Gemini).
>
> We added to the philosophical discussion on belief found in Appendix A.3, citing these other methods of obtaining belief.
>
> [1] Meng, K., et al. (2022). Locating and Editing Factual Associations in GPT. NeurIPS. https://arxiv.org/abs/2202.05262.
>
> [2] Gurnee, W., & Tegmark, M. (2024). Language Models Represent Space and Time. ICLR. https://arxiv.org/abs/2310.02207.
>
> [3] Zou, A., et al. (2023). Representation Engineering: A Top-Down Approach to AI Transparency. https://arxiv.org/abs/2310.01405.
>
> [4] Goldstein, S., & Levinstein, B. A. (2024). Does ChatGPT Have a Mind?. https://arxiv.org/pdf/2407.11015
>
> **W2: Prompt sensitivity.** We acknowledge that prompt sensitivity is a challenge in LLM evals.
> We acknowledge that prompt sensitivity is a challenge in LLM evaluations. We addressed this in the paper design through three primary mechanisms:
> 1. Ensemble approach for belief: To ensure we are not capturing an artifact of a specific belief prompt, we do not rely on a single query. As detailed in Section 4.2 and Appendix A.3, we query the model with three direct questions and two related/indirect questions. We only classify a model as holding a "belief" if it answers consistently across this ensemble. If a model’s knowledge state is sensitive to the phrasing of the belief question, it fails the consistency check and is excluded from the honesty calculation.
> 2. Pressure robustness (lying@n): To address sensitivity to generation noise and pressure strength, we included the Lying@n metric (Appendix A.8), where we evaluate responses to the pressure prompt over 10 independent generations. We found that the core trend—that larger models are more dishonest—persists even when aggregating over multiple samples (Figure 14).
> 3. Large dataset: Individual prompts can be sensitive, but MASK evaluates performance across ~1,500 examples. Furthermore, the "inverse scaling" finding across 30 different models and 6 diverse archetypes suggests that our results are driven by model behavior rather than noise from specific wording choices.
>
> **W3: Paper defines six dishonesty types but doesn’t report how many samples each contains and performance on different types.**
>
> We have added the distribution of the six archetypes in the MASK public dataset to Appendix A.7:
> * Situation-Provided Facts: 27.4%
> * Known Facts: 20.9%
> * Continuations: 17.6%
> * Disinformation Generation: 12.5%
> * Doubling Down: 12.0%
> * Fabricated Statistics: 9.6%
>
> The dataset is relatively balanced across archetypes to prevent any single scenario from dominating the score.
>
> Figure 8 already provides the breakdown of honesty by question type.
>
> **W4: Dataset availability.** We agree that it would be useful to see the dataset, and per the Reproducibility Statement, we are committed to releasing the dataset. To assist reviewers, we have made the public split of the MASK dataset (1000 examples) to an anonymous repository for review: https://anonymous.4open.science/r/MASK-Benchmark-23919

---

### Author Response · Authors · 2025-12-03
**Our comment**

We thank the reviewers for their insightful feedback and for recognizing the importance of disentangling honesty from accuracy. We are encouraged that reviewers found our concept clear, our scale and coverage strong, and our findings regarding the inverse relationship between scale and honesty significant.
We address three common themes raised by multiple reviewers below, followed by specific responses to each reviewer.

**Novelty regarding Li et al. (2024).**

Reviewers FLsH and 6iaV noted that the distinction between honesty and accuracy has been discussed in surveys like Li et al. (2025).

While we agree the theoretical distinction exists in prior literature, MASK’s contribution is the first large-scale, automated benchmark to operationalize this distinction empirically across 30 frontier models. Prior work lacked a standardized pipeline to measure lying at scale, especially for closed-weight models. MASK provides the methodology and dataset to empirically disentangle these concepts. We have revised the introduction and related work to clarify this positioning.

**Defining and Measuring "Belief" (Circularity).**

Reviewers raised important questions about our consistency-based method for eliciting beliefs. Reviewers vCtB and FLsH asked about our operationalization of "belief" and the potential for circularity (i.e., how do we know the unpressured answer is the "true" belief?).

* Operationalization: We define belief behaviorally as the model’s consistent internal state *in the absence of pressure* (using neutral prompts). We validate this by asking the belief question three times plus two related questions to ensure robustness.
* Why not probing or analyzing training data? Reviewers suggested alternatives like probing or analyzing training data. While probing is valuable, it is not possible for closed-source frontier models (GPT-4o, Claude 3.5, Gemini). Analyzing training data is also intractable for closed-source models, as training data may not be available. Our black-box approach provides a scalable method applicable to any LLM. We have expanded the discussion in Section A.2 (Limitations) regarding these trade-offs.
* Addressing circularity: We assume the "neutral" prompt elicits the model's default knowledge state (accuracy). MASK measures the deviation from this established behavioral baseline when pressure *is* applied.

We mitigate the risk of relying on potentially inaccurate or unstable baseline responses through several checks:
* Consistency: We verify robustness by querying the model multiple times (three direct queries and, for binary propositions, two related questions queries) and checking for consistency (Section 4.2, A.3). If responses are inconsistent, we classify the model as having "no belief."
* External validation: In Appendix A.5, we show that when the pressure is removed in a follow-up chat, models self-report having lied in 83.6% of the cases identified by our pipeline. This strongly corroborates that our belief elicitation method accurately captures the model's underlying knowledge, which it chose to contradict under pressure.

**Dataset Availability and Details (vCtB W3/W4).**

We agree that access to the dataset is crucial for a benchmark paper. We have uploaded the public split of the MASK dataset (1000 examples) to an anonymous repository for review: https://anonymous.4open.science/r/MASK-Benchmark-23919

Per vCtB's request (W3), we have added the distribution of the six archetypes in the MASK public dataset to Appendix A.7:
* Situation-Provided Facts: 27.4%
* Known Facts: 20.9%
* Continuations: 17.6%
* Disinformation Generation: 12.5%
* Doubling Down: 12.0%
* Fabricated Statistics: 9.6%

---

### Meta-Review · Area_Chair_LzbE · 2026-01-11

**Summary:**

All three reviewers agree the paper tackles an important safety question and that the benchmark idea is clear: separate “accuracy/knowledge” from “honesty” by eliciting a baseline belief under neutral prompts and then applying realistic pressure prompts to induce lies of commission. Reviewers also find the empirical scope strong (human-written scenarios; many frontier models) and the main result interesting: capability/scale improves accuracy but does not reliably improve honesty, and models can lie under pressure despite scoring well on truthfulness-style benchmarks.

However, the decision hinges on whether this measures “knowing deception” rather than prompt-driven variability or specifics of the belief-elicitation protocol. Two reviewers raised a methodological concern: the benchmark relies on model responses to infer “belief,” creating potential circularity (if the model can be dishonest in the belief step, how do we trust that baseline).

Reviewers also asked for stronger evidence that results are stable to prompt wording and pressure strength, plus clearer reporting of dataset composition and per-archetype performance. Reviewers also flagged gaps (lies of omission/selective framing, multilingual behavior).

This contribution is promising but not yet at the level expected for ICLR without stronger validations.

**Reviewer Concerns:**

Core construct validity: Rebuttal argues that stability across multiple neutral prompts plus a follow-up self-report (“models admit lying in 83.6% of cases”) supports the approach. This helps, but it does not fully break the circularity: both the baseline belief and the “admission” are still model-generated. For a benchmark whose main claim is “we directly measure lying,” this remains the key weakness.

Prompt sensitivity / robustness: The authors cite multi-sampling and an ensemble of belief prompts, which are good steps. But the reviewers asked specifically for paraphrase-based robustness tests and pressure-strength variation showing the qualitative conclusions are stable.

Novelty relative to prior “honesty” literature. Authors clarified that the conceptual distinction exists but claim novelty in operationalizing it at scale across closed models. That positioning is reasonable, but the paper is still perceived as incremental by two reviewers: the main novelty rests on the operationalization.

Coverage gaps: Rebuttal acknowledges this as future work. That is fine, but it remains a limitation: the benchmark covers only one important slice of dishonesty, which should be framed more narrowly in the core claims.

**Reviewer Scores:**

Reviewer vCtB: The dataset link and added archetype distribution likely address two of their concrete weaknesses. However, their major methodological concerns (belief collection and prompt sensitivity via paraphrase/pressure variation) are only partially addressed.

Reviewer FLsH: Authors responded and added larger-model LoRA results, which answers one weakness. But the circularity concern remains their core objection, and they stated they were keeping scores.

Reviewer 6iaV: Rebuttal provides a plausible hypothesis for inverse scaling and appropriately narrows omission/multilingual as limitations.

---

### Decision · Program_Chairs · 2026-01-26

Reject